# The Management of Acute Colonic Diverticulitis in the COVID-19 Era: A Scoping Review

**DOI:** 10.3390/medicina57101127

**Published:** 2021-10-18

**Authors:** Roberto Cirocchi, Riccardo Nascimbeni, Gloria Burini, Carlo Boselli, Francesco Barberini, Justin Davies, Salomone Di Saverio, Diletta Cassini, Bruno Amato, Gian Andrea Binda, Gabrio Bassotti

**Affiliations:** 1Department of Medicine and Surgery, University of Perugia, 06123 Perugia, Italy; roberto.cirocchi@unipg.it (R.C.); carlo.boselli@unipg.it (C.B.); francesco.barbenini@unipg.it (F.B.); gabassot@tin.it (G.B.); 2Department of Molecular and Translational Medicine, University of Brescia, 25121 Brescia, Italy; riccardo.nascimbeni@unibs.it; 3General & Emergency Surgical Clinic, University of Ancona, Hospital “Ospedali Riuniti di Ancona”, 60126 Ancona, Italy; 4Colorectal Unit, Addenbrooke’s Hospital, Cambridge University Hospitals NHS Foundation Trust, Hills Road, Cambridge CB2 0QQ, UK; justin.davies@addenbrookes.nhs.uk (J.D.); salo75@inwind.it (S.D.S.); 5General and Laparoscopic Surgery, ASST Nord Milano, Sesto San Giovanni, 20099 Milano, Italy; diletta.cassini@yahoo.com; 6Department of Clinical Medicine and Surgery, University of Naples “Federico II”, 80138 Naples, Italy; bramato@unina.it; 7General Surgery, Biomedical Institute, 16152 Genoa, Italy; gabinda@me.com

**Keywords:** COVID-19, SARS-CoV-2, acute diverticulitis, diverticular disease, new management, emergency room

## Abstract

*Background and Objective*: During the COVID-19 pandemic, health systems worldwide made major changes to their organization, delaying diagnosis and treatment across a broad spectrum of pathologies. Concerning surgery, there was an evident reduction in all elective and emergency activities, particularly for benign pathologies such as acute diverticulitis, for which we have identified a reduction in emergency room presentation with mild forms and an increase with more severe forms. The aim of our review was to discover new data on emergency presentation for patients with acute diverticulitis during the Covid-19 pandemic and their current management, and to define a better methodology for surgical decision-making. *Method*: We conducted a scoping review on 25 trials, analyzing five points: reduced hospital access for patients with diverticulitis, the preferred treatment for non-complicated diverticulitis, the role of CT scanning in primary evaluation and percutaneous drainage as a treatment, and changes in surgical decision-making and preferred treatment strategies for complicated diverticulitis. *Results*: We found a decrease in emergency access for patients with diverticular disease, with an increased incidence of complicated diverticulitis. The preferred treatment was conservative for non-complicated forms and in patients with COVID-related pneumonia, percutaneous drainage for abscess, or with surgery delayed or reserved for diffuse peritonitis or sepsis. *Conclusion*: During the COVID-19 pandemic we observed an increased number of complicated forms of diverticulitis, while the total number decreased, possibly due to delay in hospital or ambulatory presentation because of the fear of contracting COVID-19. We observed a greater tendency to treat these more severe forms by conservative means or drainage. When surgery was necessary, there was a preference for an open approach or a delayed operation.

## 1. Introduction

Towards the end of December 2019, the world became aware of the “Severe Acute Respiratory Syndrome Coronavirus 2” infection, caused by a Coronavirus, when the epidemic emerged for the first time in Wuhan, China and rapidly spread across the world [1]. The World Health Organization (WHO) declared the status of pandemic in March 2020. 

Patients with COVID-19 exhibit a broad spectrum of clinical symptoms: flu-like symptoms such as fever, fatigue, dry cough, shortness of breath, headache, chest tightness, chest pain, and muscle pain; however, they also present with nausea, vomiting, and diarrhea [2]. Due to the rapid and wide expansion of these patients throughout the world, this infection severely strained the various health systems and often delayed the diagnosis and treatment of other pathological conditions [3]. Concerning surgery, national health care services commonly adopted a policy of deferring all elective surgeries and performing only emergency procedures. In common clinical practice, COVID-19 infection has dramatically reduced the number of patients undergoing emergency and elective surgery, and this decrease was associated with a rise in delayed diagnoses [4].

This change is very important for some very common diseases, such as colonic diverticular disease, which includes a broad spectrum of different conditions from diverticulosis to acute diverticulitis, with a full spectrum of severity ranging from self-limiting infection to abscess or fistula formation and free perforation [5,6,7]. Similar to COVID-19 (at least in the beginning of the pandemic), the elderly are more commonly affected by diverticular disease and they tend to have a worse prognosis [8].

This raises the question of whether there is a trend during the COVID-19 outbreak toward increased frequency of late presentations to emergency departments as reported in emergency surgery for secondary peritonitis [9].

The purpose of this review was to analyze the treatment of acute diverticulitis during the COVID-19 pandemic, and to gain insights into potential pitfalls delaying the surgical care of these patients.

## 2. Methods

Scoping reviews are a method for “reconnaissance to clarify working definitions and conceptual boundaries of a topic or field” [10]. Since there is no universal scoping study definition [11], in this review the O’Malley and Arksey methodological framework was adopted: identifying the research question, searching for relevant studies, selecting studies, charting the data, collating, summarizing, and reporting the results [12].

The protocol for this study was registered on PROSPERO, an international prospective database for reviews, under the registration number CRD42020189947. The Protocol states that the study addresses COVID-19 patients hospitalized for acute complicated diverticulitis and their treatments identified with primary outcome of 30 day mortality. 

The search for relevant studies was performed on PubMed, SCOPUS and on Web of Science from March 2020 to January 2021. The PRISMA-ScR flow chart (Preferred Reporting Items for Systematic reviews and Meta-Analyses extension for Scoping Reviews) was followed as SDC 1 [13]. 

The scoping review includes patients with acute left sided colonic diverticulitis.

For definition of uncomplicated and complicated diverticulitis we have referred to Hinchey modified classification and, above all to the WSES classification [14]: uncomplicated is defined as an infection which involves only the colon, while complicated diverticulitis is defined as an infectious process extending beyond the colon and which is divided into four stages based on the extension of the infectious process:
-Uncomplicated: Diverticula, thickening of the wall, increased density of the pericolic fat-Complicated○1A. Pericolic air bubbles or small amount of pericolic fluid without abscess (within 5 cm from inflamed bowel segment)○1B. Abscess ≤ 4 cm○2A. Abscess > 4 cm○2B. Distant gas (>5 cm from inflamed bowel segment)○3. Diffuse fluid without distant free gas○4. Diffuse fluid with distant free gas

The research question of this scoping review was “what is known about COVID-19 and acute colonic diverticulitis?” The search strategy was carried out using the following keywords: “COVID-19”, “SARS-CoV-2”, “2019-nCoV”, “acute diverticulitis”, and “acute colonic diverticulitis”, in various combinations with the Boolean operators “and”, “or”, and “not”.

No language restriction was applied. When the same study group published the articles and an overlap was found, only the most recent article was included, in order to avoid data duplication. The PubMed function “related articles” was used to extend the search. We also performed a hand-search of the bibliography of included studies to identify other potentially eligible investigations.

We planned an analysis of grey literature, so we performed a search plan on google and the results were included as “other source”.

Inclusion criteria included all types of articles (review, original article, letter, position paper) on COVID-19 and acute colonic diverticulitis. All studies were independently assessed for eligibility by two reviewers (RC and GB), and eventual controversies were resolved by a consensus among the reviewers. Two authors (RC and GB) independently extracted the following data: first author name, study design, country, sample size, objective, inclusion and exclusion criteria, and outcome measures. The identified studies were summarized according to key themes based on similarities in their main intervention and metrics. Unlike the methodology for a systematic review, the risk of bias evaluation of included studies was not assessed.

For evaluation of the level of evidence of clinical recommendations in included documents, the authors rated each document using the Oxford CEBM levels of evidence [15]. No patient association/group or other stakeholders were involved in the design, conduct, reporting or dissemination plans of this scoping review.

## 3. Results

The PRISMA flow chart and checklist for systematic review is shown in Figure 1. The initial search produced 135 potentially relevant articles. After 85 articles were screened for relevance of titles and abstracts, 50 full text articles were further assessed for eligibility and 25 were excluded ( Appendix A); thus, 25 trials were included in the review [16,17,18,19,20,21,22,23,24,25,26,27,28,29,30,31,32,33,34,35,36,37,38,39,40]. Their characteristics are reported in Table 1, Table 2, Table 3 and Table 4. 

In the analysis of the included studies, it was not possible to include data on 30-day mortality or to explore the surgical and conservative treatments.


*In the “stay-at-home” period, there was a decrease in hospital access for acute diverticulitis episodes; however, the rate of associated abscess was unchanged or even increased.*


During the COVID-19 pandemic, many patients with urgent health problems avoided hospital access for clinical/radiological evaluations and treatments. For these reasons, some patients presented late with more severe conditions. An analysis performed on patients with diverticulitis who were evaluated at Virginia Mason Medical Center (Seattle, WA, USA) showed a shift to lower outpatient and increased inpatient and Emergency Department attendance (*p* < 0.01) [19]. At the same time, the incidence of associated abscesses more than doubled compared to the incidence of abscesses during non-COVID periods (11.7% vs. 4.4%, *p* < 0.01) [36], and the incidence of peritonitis (Hinchey III and IV) was significantly higher (28.8% vs. 11.2%, *p* = 0.005) [22]. Another assessment performed at the Department of Radiology of the University of Massachusetts Medical School reported a reduction of uncomplicated diverticulitis (*p* = 0.002), while there was no significant difference in the number of complicated diverticulitis cases (*p* = 0.09) [23]. The same reduction of patients with acute diverticulitis was reported in some observational studies of patients admitted at emergency surgery units in Italy [21], New Zealand [38], and the UK [22].


*The preferred treatment of acute diverticulitis (Hinchey I and II) during the COVID-19 pandemic was non-operative treatment, ideally in outpatient settings. In these patients, a telephone follow-up is needed.*


Recent recommendations and expert opinion suggest initially conservative management of acute diverticulitis with or without antibiotics, based on CT findings [23,24,25,27]. In uncomplicated and/or mild abdominal conditions (e.g., uncomplicated or Hinchey 1a diverticulitis), outpatient management and follow-up by telephone or virtually is suggested [24,27]. In cases with significant abdominal abscesses, the treatment of choice should be percutaneous drainage, reserving emergency surgery only for extreme cases (Table 5). 


*A CT scan remains the gold standard for diagnosis in cases of suspected diverticulitis, in order to distinguish between mild and complicated forms. Percutaneous drainage has assumed a primary role in diverticulitis with associated abscess formation.*


In patients with suspicion of complicated diverticulitis, expert opinion suggests performing a CT scan evaluation to distinguish the source of diffuse peritonitis in patients with or without sepsis [24,29]. According to international guidelines and statements [7,14,41,42], there is a consensus that during the COVID-19 pandemic the initial treatment of hemodynamically stable patients without diffuse peritonitis should take a non-operative approach. In selected cases of associated abscess, percutaneous drainage can reduce the need for emergency surgical treatment [24,29,30]; in the first phase of the COVID-19 pandemic, percutaneous drainage as a bridge to a surgical approach was suggested for larger abdominal abscesses [16,24,29,30].


*When surgery is requested, after an initial phase of confusion, the common opinion is now to perform laparoscopic resection with primary anastomosis, except in cases of sepsis or instability, in which open surgery is to be preferred.*


In surgical decision-making (Table 6 and Table 7), emergency surgery is suggested only in patients with diffuse peritonitis or with septic shock [29,30]. In the first phase of the COVID-19 pandemic, some experts suggested that open surgery was the preferred abdominal access and laparoscopic access should have been avoided [16,24,29,30]; and laparoscopic lavage was not recommended [24,29], nor was colonic resection with primary anastomoses [17,24,28,29]. Hartmann’s procedure was considered better than performing an anastomosis [24,29,32,33]. Concerning these last two topics, different expert opinions were reported; however, these did not consider that laparoscopic resection was to be avoided [32], as well as colonic resection, and primary anastomoses [30,31,32]. Other discordant opinions were reported on Hartmann’s procedure being better than a primary anastomosis [30]. 

In fact, the reported opinions on the previous topics are no longer accepted in common clinical practice in non-COVID-19 patients.

When surgery is requested, the expert opinions is to perform, when possible, laparoscopic resection with primary anastomosis [24], except in cases of septic shock or hemodynamic instability in which open surgery and a likely stoma are to be considered.


*In patients with COVID-related pneumonia, the preferred strategy for peritonitis from complicated diverticulitis is a non-operative treatment with the aim of delaying surgery.*


Very few studies investigated patients with COVID-related pneumonia in this setting. Traditionally, the factors in favor of non-operative treatment are Hinchey I and II diverticulitis, whereas failure of this approach favors operative treatment [35]. Costanzi et al. described a case of acute perforated diverticulitis (mesenteric free fluid-gas collection of 20 × 25 mm diameter and free intraperitoneal air) initially undergoing medical treatment and postponing surgery; however, the patient underwent emergency surgery for increased parasigmoid collection (up to 10 cm diameter). The surgical treatment was an open Hartmann’s procedure [18]. 

Now, the consensus suggests undertaking a non-operative approach with antibiotics alone when possible, and in case of abscess to consider percutaneous drainage. In this scenario, surgery must be reserved for extreme conditions, including severe sepsis and/or hemodynamically unstable patients, in which case open surgery is indicated [24,29].

Furthermore, the Italian, Spanish, and American colorectal societies agreed to indicate a conservative initial approach and to delay surgery both in patient without COVID pneumonia and in patients affected by Sars-Cov-2.

## 4. Discussion

Our scoping review aimed to analyze the changes in treatment of acute diverticulitis during the COVID-19 pandemic.

During the pandemic, worldwide health care systems collapsed; most spaces were converted into intensive care units, the medical and nursing staff were redistributed, and much of elective and outpatient activity was canceled. In addition, the urgent activities changed. Those that were affected more than others were related to surgical activity, not only elective but also acute care surgery [4]. In addition, a very important decrease in emergency room presentation was observed, likely due to the fear of infection, and an increased number of “home deaths” was observed [43,44,45].

Moreover, higher morbidity was observed in patients undergoing acute care surgery during the pandemic period, although there was no difference in mortality or reoperation rate. A frequent trend in choosing more conservative therapies in order to avoid hospitalization was also reported [45], as a decreased proportion of patients underwent a laparoscopic approach. The preference in performing an open surgical approach could be related to having more patients with a complicated course; however, an initial fear of spreading COVID-19 infection with laparoscopic aerosols could also explain this shift [45].

Since there is evidence that postoperative pulmonary complications associated with high mortality are observed in about 50% of patients with perioperative SARS-CoV-2 infection, postponing non-urgent procedures and promoting non-operative treatment to delay or avoid the need for surgery should always be accurately considered [46].

A reorganization of emergency and elective colorectal surgery pathways during the COVID-19 pandemic became necessary. For several conditions about which there was no robust evidence in the literature, we have seen some controversy concerning the approach to be adopted.

One of these diseases is colonic diverticular disease, with a variable spectrum from asymptomatic diverticulosis to acute diverticulitis up to peritonitis and abscess formation [7]. Compared to previous years, during the COVID-19 pandemic we have observed an increase in severe forms of diverticulitis in the emergency rooms [9], often with complications such as abdominal abscesses or purulent/fecal peritonitis [36,39].

Some scientific societies tried to define guidelines, especially of “good behavior” for the surgeon, to be observed during the pandemic, for both emergency and elective practice. The American College of Surgeons, the Italian Society of Colorectal Surgery, and the Spanish Association of Surgeons tried to define some guidelines to reduce the risk of COVID-19 infection in elective and urgent surgery during the pandemic. These guidelines highlighted the need for testing of all patients [47,48], the correct use of personal protective equipment, and erring towards conservative treatments in order to limit staff exposure and to limit the generation of surgical aerosol and smoke, thereby reducing the risk to patients should they contract COVID-19 in the post-operative period. This led to an initial decrease in the use of laparoscopy [23,25,27].

We have conducted a scoping review including 25 trials comprising patients with acute left sided colonic diverticulitis. We could not conduct a true systematic review due to the low levels of evidence in the small number of included cases.

We have taken into consideration five points:-The first point analyzed was the relationship between the reduced presentation to hospital during the COVID period and the presentation of more severe forms of diverticulitis, in particular those accompanied by abscess formation. However, we have found statistical significance in only one study, while in the others the incidence of complicated forms was comparable to the previous period [19].-The second aspect reviewed was the preference of treatment. Recent recommendations suggested a preferred initial conservative management with antibiotics or, in cases of abscess, percutaneous drainage, with surgery reserved only for extreme cases.-The third was, as expected, confirmation of CT scanning as the diagnostic gold standard, along with a decrease in the number of CT scans conducted but an increased number of cases of complicated diverticulitis documented [36]. We found an increase in the use of percutaneous drainage as a therapeutic procedure rather than as a bridge to surgery, and also in its use for larger abdominal abscesses [16].-The fourth point analyzes the preferred type of operation in cases of diverticulitis complicated by sepsis and when surgery was mandatory. In these cases, we found considerable confusion; at the start of the pandemic, expert guidance suggested avoiding laparoscopic approaches [16,24,30] and primary anastomoses [24], thus favoring Hartmann’s procedure. However, these recommendations are no longer accepted and expert indications are to perform, when possible, laparoscopic resection with primary anastomosis [24].

Finally, we looked to address what types of procedure is appropriate in a patient with complicated diverticulitis affected by COVID-related pneumonia; only one such report was found, in which an open Hartmann’s procedure was conducted and no indications were provided [18]. It is recognized that surgery should be avoided whenever possible in patients with active COVID-19 infection. The Italian Society of Colorectal Surgery [25], as well as the American and Spanish associations of surgeons, defined some guidelines for patients with abdominal surgical pathology; in particular, for acute complicated diverticulitis the indications should be tailored to the clinical manifestations. An initial conservative approach with observation and antibiotic treatment has been recommended. Meanwhile, in COVID-19 positive patients, open surgery may be preferred to laparoscopic surgery for Hinchey 3 and 4 patients in order to avoid aerosolized contamination.

After our review we have concluded, point by point, that:-Reduced access to emergency rooms and a large number of complicated forms of diverticulitis in consequence was found in some hospitals, but not all.-The preferred treatment of Hinchey I and II was conservative.-CT scanning in the diagnostic process and percutaneous drainage in diverticulitis complicated by an abscess increased in order to reduce surgical treatment.-Emergency surgery was suggested only in patients with diffuse peritonitis or with sepsis, and Hartmann’s procedure was preferred and recommended.-In patients with COVID-related pneumonia, the preferred strategy for peritonitis from complicated diverticulitis was non-operative treatment with the aim of delaying surgery.

This scoping review on the management of acute colonic diverticulitis during the COVID-19 pandemic attempted to analyze and summarize the current literature on the topic. The literature on this topic is of low quality, with the first reports only published in the first half of 2020. Most papers and opinions do not exceed level four evidence, and consist primarily of level five. Furthermore, some of the included papers present conflicting information, showing the development and changing of knowledge about how to work with COVID-19 during the early months of the pandemic.

For all of these reasons, we have conducted a scoping review instead of a systematic review, aware that this work is indicative and initial; however, the coming months and years will require more in-depth work, with much higher numbers in terms of patients and articles.

## 5. Conclusions

During the COVID-19 pandemic there was a clear reduction in patient presentation to emergency departments, especially for thoracic-abdominal problems, and an increase in complicated forms arriving to hospitals in poor condition. Given this scenario, we analyzed the fate of patients affected by diverticular disease and found that, while the total number of visits for diverticular disease decreased, the number of complicated forms increased. This could be related to the aforementioned late presentation in the emergency room or due to delays in seeking medical help earlier in the course of the disease, probably due to fear of contracting COVID-19. Most of these forms were treated by conservative means or with percutaneous drainage, and we noticed that in cases needing surgery there was an indicated preference for an open approach and an avoidance of anastomoses.

## Figures and Tables

**Figure 1 medicina-57-01127-f001:**
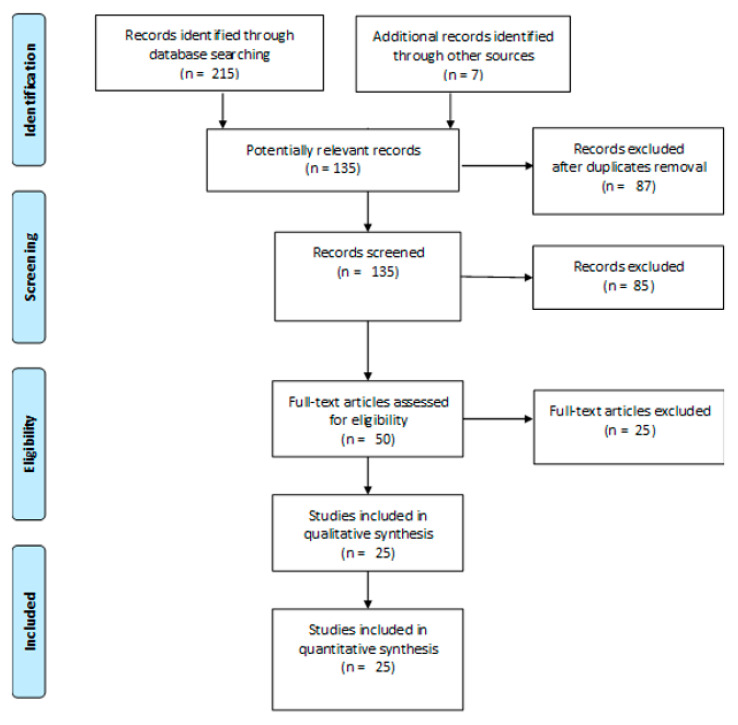
PRISMA flow diagram of study research.

**Table 1 medicina-57-01127-t001:** Included studies—Expert opinion.

Study	Nation	Year of Publication	Type of Study	Oxford Centre for Evidence-Based Medicine: Levels of Evidence (March 2009)
Mariani et al. [16]	Italy	2020	Expert opinion	5
Intercollegiate General Surgery Guidance [17]	UK	2020	Expert opinion	5
American College of Surgeons [23]	USA	2020	Expert opinion	5
Di Saverio et al. [24]	Italy	2020	Expert opinion	5
Italian society of colorectal surgery [25]	Italy	2020	Expert opinion	5
McBride et al. [26]	Australia	2020	Expert opinion	5
Asociación Española de Cirujanos [27]	Spain	2020	Expert opinion	5
Chew et al. [28]	Singapore	2020	Expert opinion	5
Di Saverio et al. [29]	Italy	2020	Expert opinion	5
De Simone et al. [30]	Italy-France–UK	2020	Expert opinion	5
Asociación Española de Coloproctología [31]	Spain	2020	Expert opinion	5
Spinelli et al. [32]	Italy	2020	Expert opinion	5
Álvarez Gallego et al. [33]	Spain	2020	Expert opinion	5
Wexner et al. [34]	USA	2020	Expert opinion	5
Parreira et al. [35]	Brazil	2020	Expert opinion	5

**Table 2 medicina-57-01127-t002:** Included studies—Individual cohort study and case report.

Study	Nation	Year of Publication	Type of Study	Oxford Centre for Evidence-Based Medicine: Levels of Evidence (March 2009)
Costanzi et al. [18]	Italy	2020	Case report	4
Soriano et al. [19]	USA	2021	Individual cohort study	2b
Gibson et al. [20]	USA	2020	Individual cohort study	2b
Rosa et al. [21]	Italy	2020	Individual cohort study	2b
Hussain et al. [22]	UK	2020	Individual cohort study	2b
Zintsmaster et al. [36]	USA	2020	Individual cohort study	2b
Patel et al. [37]	USA	2020	Case report	4
McGuinness et al. [38]	New Zealand	2021	Individual cohort study	2b
Hossain et al. [39]	UK	2020	Individual cohort study	2b
Weissman et al. [40]	USA	2020	Case report	4

**Table 3 medicina-57-01127-t003:** Difference of procedures.

	Spain	Italy	UK	Italy	Italy	Spain	Italy	Singapore	Italy	USA
	Asociación Española de Coloproctología [27]	Spinelli et al. [32]	Intercollegiate General Surgery Guidance [17]	Di Saverio et al. [24]	Di Saverio et al. [29]	Álvarez Gallego [33]	Italian society of colorectal surgery [25]	Chew et al. [28]	De Simone [30]	Wexner [34]
Published online	20 March 2020	23 March 2020	27 March 2020	31 March 2020	7 April 2020	7 April 2020	14 April 2020	29 April 2020	30 April 2020	2 May 2020
Laparoscopic resection is avoided	NR	VS	NR	5	5	NR	5	NR	5	NR
Laparoscopic lavage is avoided	NR	NR	NR	5	5	NR	NR	NR	NR	NR
After colic resection, primary anastomoses is avoided	VS	VS	5	5	5	NR	NR	5	VS	NR
The Hartmann procedure is better than the anastomosis	NR	5	NR	5	5	4	NR	NR	VS	NR
During laparoscopy, intracorporeal anastomosis is better than extracorporeal anastomosis	5	5	NR	NR	NR	NR	NR	NR	NR	NR
Open abdomen should be avoided in critical patients	NR	NR	NR	5	5	NR	NR	NR	NR	NR

VS: The Authors reported a contrary opinion.

**Table 4 medicina-57-01127-t004:** Epidemiology evaluation of acute diverticulitis during COVID-19 pandemic.

Treatment	Soriano [19]	Zintsmaster [36]	Gibson [23]	Rosa [21]	McGuinness [38]	Hussain [22]	Hossain [39]
Population	Administrative analysis of ICD10 codes (episodes of diverticulitis)	Patients evaluated with CT for AD	Patients evaluated with CT for AD	Patients admitted in emergency surgery unit	Patients admitted in emergency surgery unit	Patients admitted in emergency surgery unit	Patients evaluated with CT for AD
Results	Significantly decrease in diverticulitis episodes during pandemic (*p* = 0.004)	Diverticulitis with 11.7% of those patients presenting with an associated abscess. During the same time in 2019, many more CT studies with newly diagnosed diverticulitis were obtained, and, compared to 2020, less than half the percentage of those patients had an associated abscess (4.4%).	Decrease uncomplicated diverticulitis cases dropped significantly (*p* = 0.002) while there was no significant difference in the number of complicated diverticulitis cases (*p* = 0.09).	The prevalence of acute diverticulitis was significantly lower during pandemic (*p* = 0.004)	No difference in severity of acute diverticulitis (*p =* 0.333)	The prevalence of acute diverticulitis was significantly lower during pandemic (*p* < 0.05)	During the COVID-19 pandemic, fewer patients presented and were diagnosed with acute diverticulitis (decrease of 51.4% than same period in 2019). A significantly greater proportion presented at a more advanced stage and required emergency surgery, suggesting late presentation

**Table 5 medicina-57-01127-t005:** Preferred strategy for uncomplicated acute diverticulitis during COVID-19 pandemic.

	American College of Surgeons [23]	Di Saverio et al. [24]	Italian Society of Colorectal Surgery [28]	McBride et al. [26]	Asociación Española de Cirujanos [31]
Non Operative Management is preferred	5	5	5	5	5
Outpatient management is preferred	NR	5	NR	NR	5
Telephone follow-up is preferred	NR	5	NR	NR	5

**Table 6 medicina-57-01127-t006:** Decision making for complicated diverticulitis during the COVID-19 pandemic.

	American College of Surgeons [23]	Chew et al. [28]	Di Saverio et al. [24]	Di Saverio et al. [29]	De Simone [30]	Mariani et al. [16]
Pre-operative CT scan	NR	NR	NR	5	NR	NR
Distinguish patients with or without diffuse peritonitis	NR	NR	NR	5	NR	NR
Distinguish patients with or without sepsis	NR	NR	NR	5	NR	NR
Initial non-operative approach in stable patients without diffuse peritonitis	NR	NR	5	5	5	4
Percutaneous drainage to reduce emergency surgical treatment	5	5	5	5	5	4
Emergent surgery in patients with diffuse peritonitis or with sepsis	NR	NR	NR	5	5	NR
Open surgery is the best abdominal access	NR	NR	5	5	5	4
The aim of the surgical treatment is the source control of infection	NR	NR	5	5	NR	NR

**Table 7 medicina-57-01127-t007:** Decision making in surgical treatment for complicated colonic diverticulitis during the COVID-19 pandemic.

	Asociación Española de Coloproctología [27]	Spinelli et al. [32]	Intercollegiate General Surgery Guidance [17]	Di Saverio et al. [24]	Di Saverio et al. [29]	Álvarez Gallego [33]	Italian Society of Colorectal Surgery [25]	Chew et al. [28]	De Simone [30]	Wexner [34]
Laparoscopic resection is avoided	NR	VS	NR	5	5	NR	5	NR	5	NR
Laparoscopic lavage is avoided	NR	NR	NR	5	5	NR	NR	NR	NR	NR
After colic resection, primary anastomoses is avoided	VS	VS	5	5	5	NR	NR	5	VS	NR
The Hartmann procedure is better than the anastomosis	NR	5	NR	5	5	4	NR	NR	VS	NR
During laparoscopy, intracorporeal anastomosis is better than extracorporeal anastomosis	5	5	NR	NR	NR	NR	NR	NR	NR	NR
Open abdomen should be avoided in critical patients	NR	NR	NR	5	5	NR	NR	NR	NR	NR

VS: The Authors reported a contrary opinion.

## Data Availability

All data are available on PubMed, SCOPUS and on Web of Science and are reported in the attached tables.

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
