# Peer review of "The Management of Acute Colonic Diverticulitis in the COVID-19 Era: A Scoping Review"

_medicina, 2021, doi:10.3390/medicina57101127_

Round 1

Reviewer 1 Report

This is a scoping review article that aimed to evaluate 5 points in regards to diagnosis and management of diverticular disease (DD) in relation to restricted resources due to COVID 19 pandemic. I have the following comments and questions I would ask authors to add/clarify in order to improve the quality of the paper. 

--In Introduction section, line 54- it should be added that similar to COVID 19 ( at least in the beginning of the pandemic) , elderly are  particularly commonly affected  by DD and they tend to have worse prognosis ( see the following : https://pubmed.ncbi.nlm.nih.gov/30792972/

--Methodology section/ PRISMA diagram: 1. What are the "other sources"?. 2. "records excluded 85 and 25"- why they were excluded? This has to be specifically described; 3. Did you include case reports and case studies? 

--Methodology- please define complicated DD vs non complicated. Please see reference above

--What was the reason to select only patients with left sided DD?

--Results: Did author document what " conservative management means"? What was the choice of antimicrobial agents? 

--Results: It would be important to include mean/median age of the patients in these studies and see if this change has any predilection for younger/older age? 

--Discussion- Should be shortened, especially the first to paragraphs are repetitive from introduction

--Discussion- one could think that if patients delayed presentation to ER due to COVID related reasons, incidence of uncomplicated forms of DD would decrease, but complicated forms would increase ( due to delayed treatment for noncomplicated cases); It would be important for authors to describe this

Reviewer 2 Report

Dear Author’s 

I have reviewed your manuscript.
Your article is very intreasting,  and  I think that a revision coud improve  your article.

1. There are 2 intreasting papers that will be very informative if you coud cite

Bellato V, Konishi T, Pellino G, An Y, Piciocchi A, Sensi B, Siragusa L, Khanna K, Pirozzi BM, Franceschilli M, Campanelli M, Efetov S, Sica GS; S-COVID Collaborative Group. Impact of asymptomatic COVID-19 patients in global surgical practice during the COVID-19 pandemic. Br J Surg. 2020 Sep;107(10):e364-e365. doi: 10.1002/bjs.11800. Epub 2020 Aug 6. PMID: 32767367; PMCID: PMC7929295.

Bellato V, Konishi T, Pellino G, An Y, Piciocchi A, Sensi B, Siragusa L, Khanna K, Pirozzi BM, Franceschilli M, Campanelli M, Efetov S, Sica GS; S-COVID Collaborative Group. Screening policies, preventive measures and in-hospital infection of COVID-19 in global surgical practices. J Glob Health. 2020 Dec;10(2):020507. doi: 10.7189/jogh.10.020507. PMID: 33110590; PMCID: PMC7567431.

2. The are minor errors in the section references. Please revise.

3.  Please explaine the novelty of your study.

Reviewer 3 Report

In this scoping review Cirocchi et al showed that during COVID19 pandemic, the rate of emergency room accesses due to diverticular disease decreased. Similarly, a trend in favour of conservative treatment of acute diverticulitis was noted.

Authors analyzed both observational papers and expert opinion articles- such papers were mixed in table 1, and I feel that this causes misunderstanding and confusion. Please split into two tables.

Author Response

I attach the cover letter with all 3 revisions due to a mistake in the response before. I'm sorry. I communicate the fact to editors. I hope it could be possible to correct. By the way all the change requested can find in the manuscript revised.

Thack you very much

Round 2

Reviewer 1 Report

The authors have successfully responded to my comments. The manuscript has been improved. I would suggest the following in order to further improve the manuscript:

  1. English should be improved.
  2. WSES classification should be added to methodology regarding definition of uncomplicated vs complicated DD
  3. I would add the sentence in the abstract and in conclusion that clearly describe the most important finding. Something like " While the total number of visits for DD has decreased during covid time, the number of complicated forms have increased. This is likely related to late presentation or due to delay in seeking medical help earlier in the course of disease, possibly due to the fear of contracting COVID 19". 

Author Response

Response to Reviewer 1 Comments

POINT 1. English should be improved.

RESPONSE:

Thank you for your suggestion. We have send the manuscript to our native language co-author Justin Davies in the UK, who provided to correct the manuscript.

POINT 2. WSES classification should be added to methodology regarding definition of uncomplicated vs complicated DD

RESPONSE

Thank you for the suggestion. We put the classification in the methods paragraph, and we cited it, so you can see a modification in the references.

POINT 3. I would add the sentence in the abstract and in conclusion that clearly describe the most important finding. Something like " While the total number of visits for DD has decreased during covid time, the number of complicated forms have increased. This is likely related to late presentation or due to delay in seeking medical help earlier in the course of disease, possibly due to the fear of contracting COVID 19"

RESPONSE

Thank you very much. We welcomed your advice and included the diction both in the abstract and in the conclusions

Reviewer 2 Report

Dear Author’s 

I have reviewed the revised version of your manuscript and i think that your article is ready to be published.

Author Response

POINT 1. I have reviewed the revised version of your manuscript and i think that your article is ready to be published

RESPONSE:

I’m very pleasure to read this. I want to thank you very very much to review our article, to have put your time and attention on it and to have approve it. Really thank.

By the way, if you want we have upload a more revised versione. Thank you again

Best regards.